# Cancer Cell Direct Bioprinting: A Focused Review

**DOI:** 10.3390/mi12070764

**Published:** 2021-06-28

**Authors:** David Angelats Lobo, Paola Ginestra, Elisabetta Ceretti, Teresa Puig Miquel, Joaquim Ciurana

**Affiliations:** 1Department of Mechanical and Industrial Engineering, University of Brescia, V. Branze 38, 25123 Brescia, Italy; d.angelatslobo@unibs.it (D.A.L.); elisabetta.ceretti@unibs.it (E.C.); 2New Therapeutic Targets Laboratory (TargetsLab), Oncology Unit, Department of Medical Sciences, Girona Institute for Biomedical Research, University of Girona, Emili Grahit 77, 17003 Girona, Spain; teresa.puig@udg.edu; 3Product, Process and Production Engineering Research Group (GREP), Department of Mechanical Engineering and Industrial Construction, University of Girona, Maria Aurèlia Capmany 61, 17003 Girona, Spain; quim.ciurana@udg.edu

**Keywords:** bioprinting, 3D printing, cancer

## Abstract

Three-dimensional printing technologies allow for the fabrication of complex parts with accurate geometry and less production time. When applied to biomedical applications, two different approaches, known as direct or indirect bioprinting, may be performed. The classical way is to print a support structure, the scaffold, and then culture the cells. Due to the low efficiency of this method, direct bioprinting has been proposed, with or without the use of scaffolds. Scaffolds are the most common technology to culture cells, but bioassembly of cells may be an interesting methodology to mimic the native microenvironment, the extracellular matrix, where the cells interact between themselves. The purpose of this review is to give an updated report about the materials, the bioprinting technologies, and the cells used in cancer research for breast, brain, lung, liver, reproductive, gastric, skin, and bladder associated cancers, to help the development of possible treatments to lower the mortality rates, increasing the effectiveness of guided therapies. This work introduces direct bioprinting to be considered as a key factor above the main tissue engineering technologies.

## 1. Introduction

Three-dimensional (3D) printing, also known as Additive Manufacturing (AM), has its origin in 1986, with stereolithography (SLA) as the first 3D printing technology [1,2]. Stereolithography uses photosensitive materials to be solidified or cross-linked by an ultraviolet light source [3] and other light sources. Afterward, other 3D printing technologies were developed, such as extrusion-based printing, inkjet printing [4], selective laser melting [5], and selective laser sintering [6]. Typically, the engineering and biology fields worked separately, however, in the early 2000s, a new field known as 3D bioprinting was introduced by Thomas Boland’s group at Clemson University [7]. One of the objectives of 3D bioprinting is to produce 3D constructs, called scaffolds, to mimic the native microenvironment, thus avoiding the ethical problems of animal experimentation [8,9].

Although different technologies may be used in 3D bioprinting, the most popular technology is extrusion-based printing [10]. This technology admits a wide range of materials, alone or combined with cells, in the form of a bioink [4]. In tissue engineering scaffolds, the process is organized in two steps, corresponding to the printing of the scaffold and the culture with cells. Mainly in extrusion bioprinting, these processes can be performed simultaneously in direct or one-step bioprinting, or separately in indirect or two-step bioprinting [11]. Direct bioprinting is highly efficient due to performing the two stages at the same time, which could be a way to increase cell integration and viability within the scaffold and, therefore, accurately mimic the native extracellular matrices. Other fields like engineering, healthcare, and research use 3D printing as the manufacturing process. Because of that, this paper will focus on research and, specifically, the use of direct bioprinting as an alternative to conventional 3D printing or indirect bioprinting, in the topic of cancer research.

Recently, in 2020, cancer caused approximately 10 million deaths worldwide [12]. A notable increase in the cost of available treatments due to relapses, morbidity, and patient time [13], may also increase the incidence and mortality rate of breast cancer worldwide. One of the theories that explains drug resistance and recurrence of the tumors suggests that within the tumor there is a subpopulation of cancer cells with self-renewal capabilities to maintain the tumor growth, proliferation, and differentiation. These are called cancer stem cells (CSC) [14]. A wide range of materials and 3D printing technologies might be used depending on the properties of the materials themselves, the specific requirements for each 3D printing technology, and the final application of the study.

As previously stated, direct bioprinting uses bioinks by the combination of biomaterials and cells. Depending on the nature of the applications, the materials of the bioinks can be decellularized extracellular matrices (dECM), tissue spheroids, cell pellets, and hydrogels [15]. The hydrogels have attractive uses in bioprinting due to their solid/aqueous state, temperature and humidity dependence [16], biodegradability, biocompatibility, adaptive mechanical strength, and availability [10]. Their limitations of dissolution kinetics in body fluids and problems in the sterilization process can be avoided with the substitution of other materials, such as metals, ceramics, and polymers, depending on the final purposes of the study [11]. Depending on the applied technology and application, the ideal properties of each bioink may vary. For example, in extrusion printing, the bioink must be biocompatible and with specific viscosity [17,18], while maintaining cell viability and functionality after the bioprinting processes [19].

## 2. Materials

In native tissues of multicellular organisms, like humans, cells secrete structural and functional molecules to dynamically maintain the extracellular matrices [20]. Extracellular matrices sustain the structural integration of tissues and contain several chemical signals to maintain cell survival, proliferation, organization, and differentiation [21]. The matrix composition can be imitated by using natural or synthetic polymers, depending on the final purpose. Ashby et al. used a selection system divided into three stages known as the initial screening, alternative postulations, and final decision of the best material for each application [22], which might be adapted to select the biomaterials for direct bioprinting demands. This adapted methodology uses property limits, geometric restrictions, material indexes, and the cost and performance of the materials, to better select the optimized bioink for each case. In vivo, the extracellular matrices (ECMs) are in constant remodeling and store bioactive molecules involved in the regulation of internal processes, such as angiogenesis. Moreover, ECMs have physical properties, such as rigidity, density, porosity, insolubility, and spatial orientation, that could be reproduced by using different natural and synthetic materials. The extracellular matrices are mainly composed of collagen, proteoglycans with glycosaminoglycans (GAGs), such as hyaluronic acid, elastin, cell-binding glycoproteins [23], and other cell adhesion peptides, such as RGDS motifs, which can be employed as natural polymers when performing the bioprinting process, for example, in the form of bioinks. Also, the bioinks must be optimized according to the typical 3D printing parameters of reproducibility, structural stability, and fidelity, as well as not being cytotoxic, with controlled degradability, compatible with cell attachment, porous [19,24], and stable during sterilization procedures [25]. Also, the use of non-Newtonian fluids is very interesting in bioprinting due to their thixotropic effects. In other words, when those fluids are exposed to more stress or external forces, they tend to reduce their viscosity [26]. As previously mentioned, controlled degradability is interesting in bioprinting to avoid pro-inflammatory responses [27,28]. Furthermore, the mechanical forces within the extracellular matrix are important for the regulation of cellular functions [29,30], such as apoptosis [31], differentiation [32], RNA processing [33], and gene expression [34].

As for the selection of articles, they were selected using databases like PubMed, Google Scholar, and others, using specific criteria around the topic of cancer research and bioprinting. The information has been organized into three main categories, corresponding to materials, 3D printing types and techniques, and specific applications using cancer cells. The last section gives a special focus on direct or one-step bioprinting, using different types of cancer cells. The methodology used can be seen in Figure 1.

### 2.1. Natural-Derived Biomaterials

Natural biomaterials have interesting properties related to 3D bioprinting, such as great biocompatibility and biodegradability, among others. These properties make them interesting for 3D bioprinting applications, from modeling to therapeutic uses.

In the literature reviewed, the main natural polymer used is collagen, which is also the most abundant component in the extracellular matrices [35]. This material can be used alone or combined with other materials, such as gelatin [36] and chitosan [37], among others. The majority of the articles use naturally derived biopolymers (71.83%) compared to synthetic polymers (28.17%). Thus, there is a tendency for using naturally derived materials instead of synthetic ones. Even though natural polymers are widely used, they also have certain limitations, for example, weak structure and poor mechanical properties. For example, Hermida et al., added RGD peptides, hyaluronic acid, and collagen type I, to improve cell adhesion of alginate structures [38]. Gelatin and chemical modifications such as gelatin methacrylate, have been grouped in the category of “collagen and derivatives”. The same strategy has been employed for the “alginate and derivatives” category.

In Figure 2, a graphic explanation of the different natural and naturally derived biomaterials used in bioprinting is reported.

As seen in Figure 2, all the materials have been organized into three groups, depending on their frequency of use.

For example, collagen and alginate derivatives point up to the most frequent materials used in cancer bioprinting. Using the concept of tumor-on-a-chip, Yi et al. used collagen combined with a glioma cell line (U-118) and endothelial cells (HUVEC), as a way to create a model for the study of glioblastoma [39].

Wang et al. employed the same glioma cell line but used a scaffold composed of gelatin, alginate, and fibrinogen, achieving a good cost–performance ratio [40].

Less frequent materials, such as Matrigel and silk, have been used in combination with collagen and a breast cell line (MCF-10A) to produce a breast model to study the differences between normal and tumorigenic breast processes [41].

### 2.2. Synthetic Polymers

In contrast, synthetic polymers have better reproducibility due to more controlled chemical manufacturing. The reduced reproducibility of natural materials might be explained by the effect of batch-to-batch variability typical of naturally derived sources. When there is a variation in the material used, even from the same brand, this can cause some errors when performing the same bioprinting process [42]. Because synthetic polymers have a low contribution (28.37%) in the literature reviewed, the different types of polymers have been organized according to their source or origin. As reported in Figure 3, the main synthetic polymers derive from polyethylene glycol (PEG), such as Pluronic F-127, used as a sacrificial material to study the photothermal treatments and tissue regeneration in bone cancer [43]. Gill et al. used PEG with RGDS peptides (PEG-RGDS) and PEG-modified with matrix metalloproteinase sensitive (PEG-PQ) scaffolds to study the process of epithelial-mesenchymal transition of lung cancer [44].

Jeon et al. used an alternative approach, using microfluidics theory [45] and polydimethylsiloxane (PDMS) to study, on a tumor-on-a-chip concept, the specificity of breast cancer metastasis sites in bone cancer [46]. Yang and Zhao demonstrated the use of a synthetic peptide, RADA16-I, as a nanofiber scaffold for anticancer drug testing and ovarian tumorigenic studies [47].

To sum up, in Table 1, there is a description of the materials mentioned above, with information of their source and the advantages and disadvantages of using them for bioprinting applications.

## 3. 3D Printing Techniques

3D bioprinting was proposed as a novel method to fill the gap between preclinical and clinical studies [77]. The requirements of the materials may vary between different techniques. For example, in micro-extrusion, the best printable biomaterials are those with high viscosity [78].

In all the literature reviewed, the majority of the articles use extrusion-based printing techniques. The other techniques are laser printing, lithography, inkjet bioprinting, droplet-based bioprinting, electrospinning, gas foaming, and freeze-casting method. Heinrich et al. applied the extrusion technology to study the cell interactions and possible therapeutic uses in glioblastoma cancer research [79]. Other authors, such as Vinson et al., used a laser printing technology called laser direct writing (LDW) to study cancer invasion in adipose tissues [80]. Stereolithography has been used by Chen et al. to evaluate the capture efficiency of cancer circulating tumor cells (CTC), which could be applied as biomarkers for early detection of cancer [81,82]. Inkjet and droplet-based bioprinting have been used to model cancer cell kinetics [83] and to study the epithelial-mesenchymal transition (EMT) related to metastatic processes on lung cancer [44]. Electrospinning, gas foaming, and freeze-casting methods have mainly been described as indirect bioprinting techniques to study the cancer microenvironments or niches [84], to model angiogenesis and metastasis in brain cancer [37], and to study oral, lung, breast, and glioblastoma malignancies [85].

In a general view, only 34.67% of the articles dealing with 3D bioprinting used an indirect bioprinting methodology, possibly because, for some research groups, it is cheaper than direct bioprinting. The rest of the articles used direct bioprinting (65.33%), mainly extrusion-based technologies, laser printing, inkjet bioprinting, droplet-based bioprinting, and a small proportion of lithography techniques. In Table 2, there is a description of the main materials used, the cost, speed, and general problems of each type of 3D printing technology.

To avoid any misunderstandings, two terminologies will be used in the topic of 3D printing technologies. On the one hand, bioprinting uses scaffold structures that could be printed and then cultured with cells, in the case of indirect bioprinting, or printed with the cells using a bioink, in the case of direct bioprinting. On the other hand, bioassembly does not use a scaffold or support material but rather is the direct bioprinting of cells, forcing their self-organization [91].

### 3.1. Bioprinting Methodologies

The extracellular matrix (ECM) has a complex, unique, and tissue-specific organization with structural and functional compounds, disposed of in three dimensions. In the early 1990s, a theory in tissue engineering was proposed, as a system to produce a biological substitute that mimics some of the functions of the ECM [92]. As mentioned earlier, more than 50% of the market is focused on the use of extrusion-based printers for bioprinting applications [10], like tissue engineering, disease modeling, and therapies testing. The extrusion printers can use pneumatic, mechanical, or electromagnetic forces to perform the actual 3D bioprinting process [93].

As previously mentioned, scaffolds can be manufactured by an arrangement of different materials, natural or synthetic, to achieve the goal of mimicking the extracellular matrices. In tissue engineering, the scaffolds contribute as mechanical supports and control the stresses generated as an artificial matrix for cell culture [94]. The scaffolds manufactured can move from a matrix [95] to a more complex system, such as a microfluidic device [96].

### 3.2. Bioassembly Methodologies

On the other hand, bioassembly needs higher cell densities, producing a more realistic extracellular matrix and self-assembly of cells that occurs in vivo [97]. The absence of physical barriers, the scaffolds, and the lack of pro-inflammatory materials improves the ECM deposition, remodeling, and integration after implantation. Related approaches, such as cell sheet technology and classic cell suspension injection, are based on extrusion printing technologies characterized by low resolution and accuracy, but high cell-to-cell communication, cell viability, and affordability [98]. Spheroids might solve these limitations by providing a physicochemical environment more similar to the native tissues. These aggregates suppress the limitations of traditional or monolayer cultures, by enabling cell-to-cell and cell-to-matrix communications in the micro-scale [99]. Several methods may be used to produce spheroids, such as hanging drop, gel embedding, magnetic levitation, and spinner culture.

The hanging drop technique produces controlled size spheroids, using surface tension and gravitation force, in droplets [100]. Alternatively, multichannel pipetting generates a high quantity of spheroids without expensive requirements [101]. Han et al. used that technique to optimize the spheroid [102], while Yip and Cho used that as an anti-cancer drug development platform [103].

Van Pel et al. combined the extrusion printing with spheroids from glioblastoma cell lines, to model the glioma invasion process that occurs in vivo [104]. On the other hand, inkjet printers can also bioprint cell suspensions to develop a breast cancer model for drug discovery and testing [105].

## 4. Cellular Classification for Direct Bioprinting

From now on, all the examples will be related to direct bioprinting, with or without using scaffolds. The different cancer cells have been organized in categories, structured in descending order depending on the up-to-date literature, the nowadays impact of the disease, and the availability of the information about cancer cells used for tissue-engineered substrates.

### 4.1. Breast Cancer Cells

Breast cancer is the second leading cause of death in women worldwide [12]. In this section, detailed information on the composition of the hydrogels or bioinks, the type of breast cancer cells used, and the 3D direct bioprinting technology employed are reported (Table 3). For example, Reid et al. used a customized extrusion printer and collagen scaffolds to develop a new theory on how the microenvironment interacts with breast cancer cells [106]. Alternatively, Kingsley et al. improved cell encapsulation with laser direct writing technology, with a wide range of applications, such as mass production of microbeads, tissue engineering, and drug kinetics [107]. Only Han et al. used bioassembly through spheroids to improve their manufacturing process [102]. The breast cancer cells in Table 3 are marked in black. In breast cancer, alginate, collagen, and derivatives are the most common materials used, with a general predominance on natural materials (81.82%) instead of synthetic ones (18.18%). The main 3D printing technologies employed are extrusion (61.11%) and laser printing (16.67%). As for the cell lines used, the majority of the articles use MDA-MB-231 and MCF-7 (75.86% of the articles), two immortalized breast cancer cell lines.

### 4.2. Brain-Associated Cancer Cells

Brain cancer is defined as a heterogeneous group of tumors derived from cells within the central nervous system. About 75% of malignant primary brain tumors are gliomas [117]. According to pre-molecular data from the WHO organization, in the United States (USA), brain cancer is the deadliest type of cancer, with less than 35% of patients surviving five years. The research should be focused on the study of gliomas and the translation of all the information on finding an effective treatment for those patients, increasing their survival rate, and reducing the tendency of relapses and death. In Table 4, detailed information on the bioinks composition, type of brain and brain-associated cancer cells used, and 3D direct bioprinting technology used are reported. For example, a novel technique known as coaxial extrusion printing, mimicked the natural drug resistance of cancer cells, allowing a better understanding for anticancer drug development [118]. The bioinks were fabricated by a combination of different naturally derived materials, except Zhang et al., who combined a microfluidic device with inkjet printing to study anticancer drug metabolism and diffusion [119], and Van Pel et al., who used a bioassembly method to model glioma invasion [104]. The brain-associated cancer cells are marked in black. In brain cancer, collagen and derivatives are the main materials used (39.29%), with a general predominance on natural materials (92.86%). The main 3D printing technology employed is extrusion (83.33%), followed by inkjet and droplet printing (8.33% each). The majority of the articles use glioblastoma cell lines (55.56%) and glioma stem cells (27.78%) for their applications.

### 4.3. Lung-Associated Cancer Cells

According to the WHO organization, in 2020, 1.8 million people died due to lung cancer. Some environmental and genetic factors may increase its frequency, such as active and passive smoking [125], asbestos [126], radon, chromium, nickel, polycyclic aromatic hydrocarbons, inorganic arsenic compounds, and bis-(chloromethyl) ether, and alpha-1-antitrypsin deficiency allele [127]. According to Doll and Peto’s study, quitting smoking for 20 years reduced cancer mortality concerning smoking [128]. Taking into consideration all the data, the research on lung cancer should be focused on modeling the disease and developing new targets. In Table 5, detailed information on the bioinks composition, type of brain- and lung-associated cancer cells used, and 3D direct bioprinting technology used is shown. For example, Wang et al. demonstrated the importance of 3D printing technology, to mimic the native lung cancer microenvironment [129]. Polyethylene glycol-derived materials have been used with patient-derived lung cancer cells, to study the epithelial-mesenchymal transition (EMT) naturally occurring in lung cancer [44] and to study the influence of vascularization on tumor progression [130]. The lung-associated cancer cells are marked in black. In lung cancer, polyethylene glycol derivates are the main materials used (45.45%), with a little bit more prevalence of synthetic materials (54.55%) instead of natural ones. Only two 3D printing technologies are employed, extrusion (75%) and droplet (25%) printing. As for the cells used, the majority are derived from patients (57.14%) instead of being immortalized cell lines (42.86%).

### 4.4. Liver-Associated Cancer Cells

Liver cancer affects global health challenges and is growing worldwide [132]. According to an estimation, by 2025, approximately more than 1 million individuals will be affected by liver cancer [133]. Several factors may increase the risk of developing liver cancer, such as viral infections of hepatitis B and C [134], exposure to aristolochic acid present in some Asian natural treatments [135], tobacco [136], and non-alcoholic steatohepatitis (NASH) [137]. Table 6 reports detailed information on the bioinks composition, type of liver-associated cancer cells used, and 3D direct bioprinting technology used. For example, Xu et al. employed encapsulated liver cancer cells to study the metastasis in vitro [138]. Also, using liver cancer spheroids, Yip and Cho demonstrated the possible use of spheroids as an alternative drug testing method [103]. The liver-associated cancer cells are marked in black. In liver cancer, the main material used is alginate (33.33%), with an equal prevalence of both natural and synthetic polymers. The main 3D printing technology is hanging drop (50%) used in bioassembly [102] and bioprinting [103] applications. The main cell line used, HepG2, came from a young patient. As for the other two cell lines, MHCC97L and HCCLM3, are derived from adult patients.

### 4.5. Reproductive-Associated Cancer Cells

The predisposition to cancer may be transmitted to the offspring/descendants, so it is very important to identify the cancer predisposition genes (CPGs), like tumor suppressor genes discovered in retinoblastoma cases [139,140]. Some examples of reproductive-related cancers may be ovarian and cervical cancers.

Ovarian cancers can be classified into different histological subtypes, such as serous, endometroid, clear-cell, and mucinous carcinomas. Women diagnosed with an advanced stage may develop resistance to platinum-based chemotherapy, complicating their survival [141]. Other risk factors may be mutations on genes involved in DNA reparation such as BRCA1 and BRCA2 [142], Lynch syndrome [143], administration of oral contraceptives [144,145], surgeries on reproductive systems [146,147], obesity [148], and smoking [149].

On the other hand, cervical cancer is commonly caused by high-risk subtypes of human papillomavirus (HPV) and might be avoided by HPV screening and vaccination programs [150]. The most common subtypes are squamous carcinomas and adenocarcinomas [151]. Early detection and possible complications may occur due to immunosuppression events, like in the human immunodeficiency virus (HIV) treatments [152,153].

Table 7 shows detailed information on the bioinks composition, type of ovarian, cervical, and germline cancer cells used, and 3D direct bioprinting technology used. For example, Ringeisen et al. proposed laser direct printing as an alternative 3D bioprinting technology to study heterogenic 3D cancer microenvironments [154]. Yang and Zhao used a synthetic peptide, RADA16-I, with ovarian serous and endometrioid subtypes, to verify the alternative use of peptide scaffolds for drug trials and tumor studies [47]. The cancer cells are marked in black. In reproductive-associated cancers, the main material used is Matrigel (28.57%), with a clear predominance of natural materials (85.71%) instead of synthetic ones. The main 3D printing technology applied is extrusion printing (80%), with only one case using laser direct writing with an embryonal carcinoma cell line [154]. Ovarian cancer is the most studied, corresponding to 71.43% of the articles, followed by cervical and embryonal cancers.

### 4.6. Gastric and Colorectal Cancer Cells

Gastric cancer has poor survival rates worldwide, and is commonly detected in Asia and South American countries [158]. Depending on the cellular components of the disease, gastric cancer can be classified as a well differentiated, poorly differentiated, or mixed disease [159]. The well differentiated disease is predominant in males over 70 years, with large tumors [160]. Poorly differentiated patients are mainly young women, with poor survival and terrible early detection [161]. The mixed disease is less frequent, usually in males, and highly invasive and metastatic [162,163]. Several predisposing factors can be described, such as pathogenic infections by Helicobacter pylori or Epstein Barr virus [164,165], genetic inheritance, and environmental effects.

On the other hand, colorectal cancer is the fourth most deadly cancer worldwide, representing 10% of all annually diagnosed cancers worldwide [133]. Colorectal cancer rising is related to aging, family history [166], medical history of long-standing inflammatory bowel disease, and previous colorectal cancers or adenomas [167,168].

In Table 8, detailed information on the bioinks composition, type of gastric and colorectal cancer cells used, and 3D direct bioprinting technology used are reported. Alginate-based bioinks have been used to study curcumin anticancer effect on colorectal cancer [169] and the influence of hyaluronic acid on gastric cancer stem cells [170]. The cancer cells are marked in black. In gastric and colorectal cancers, the main material used is alginate, used in combination with extrusion printing. In Table 8, there is only one case for colorectal cancer and gastric cancer research, in that order.

### 4.7. Skin-Associated Cancer Cells

Skin cancer is the most commonly diagnosed cancer in Caucasians [171,172,173], classified as malignant melanoma (MM) and non-melanoma skin cancer (NMSC). In recent years, the incidence of both types of skin cancer has increased [174], with the NMSC between 18 to 20 times higher than MM [175,176]. One explanation of the higher incidence of NMSC, especially in the basal cell carcinoma (BCC) and the squamous cell carcinoma (SCC) subtypes, is the overuse of recreational UV, such as indoor tanning [177,178].

In Table 9, detailed information on the bioinks composition, type of skin cancer cells used, and 3D direct bioprinting technology used is provided. For example, both studies focus on how the composition of the hydrogels influences the cancer phenotypes observed, also remarking the importance to select the best bioink for each application [116,179]. The cancer cells are marked in black. In skin cancer, two strategies have been employed to test different hydrogels and bioinks to model melanoma cancer [116] and to study the influence of the composition of the hydrogels on the cancer cell phenotypes [179], using extrusion printing, with a general predominance of natural materials (80%).

### 4.8. Urinary Bladder Cancer

Bladder cancer is the ninth most common cancer worldwide [180]. About 75% of bladder cancers are superficial or non-muscular invasive types [181]. Age, particularly between 70 and 84 years, has been reported as a high-risk factor for bladder cancer, due to exposure to carcinogens and the reduction of efficacy of DNA repair systems [182]. Also, men are three to four times more predisposed to develop bladder cancer than women, but poor diagnosis in women due to confusion on hematuria development (blood in urine) may reduce their survival [183]. Other factors such as chronic inflammation [184] and pelvic radiations [185,186] may also increase that risk.

Kim et al. used a scaffold composed of ultraviolet (UV) cross-linkable gelatin methacrylate for anticancer drug testing [187].

## 5. Conclusions

3D cancer bioprinting is mainly based on extrusion for the bioprinting of scaffolds, even though bioassembly and other technologies, such as laser direct writing, may also be used. Natural and synthetic materials, mainly collagen-derived and PEG-derived compounds, are used for several applications, primarily cancer disease modeling. Because these studies try to emulate the natural extracellular matrix and the microenvironments, different combinations of patient-derived cells and immortalized cell lines have been used, which reduces the instant need for animal experimentation in pre-clinical studies. Cancer research studies have been organized according to the level of knowledge of each type of cancer, with breast cancer the most studied using direct bioprinting. This can be very alarming because some cancers, such as liver cancer, are expected to increase by 2025. Compared with breast cancer, gastric-related and skin cancers have a low contribution on cancer research using direct bioprinting, which is a little bit strange, because one common metastatic site for melanoma is the gastrointestinal tract [188]. For urinary bladder cancer, the research is mainly focused on bladder replacement, for example, using collagen and polyglycolide scaffolds cultured with autologous bladder urothelial and muscle cells [189], or generation of organoids using transurethral or xenograft resections [190]. In colorectal cancer, the main research is focused on the establishment of an in vitro 3D model, using colorectal cancer cells (HCT 116) with collagen and polycaprolactone scaffolds combined with animal experimentation [191] or employing an encapsulator machine for alginate microbead casting [192]. The large contribution of breast cancer research may be explained as a result of being the second main cause of death in women [12]. Recent advances such as bioassembly methods [97] and material science may improve the understanding of how materials influence cell-to-cell and cell-to-scaffold interactions. Using novel techniques such as the tumor-on-a-chip to study patient-specific glioblastomas [122], collagen-hydroxyapatite scaffolds as osteochondral substitutes [193], and the optimization of the proper bioprinting process [194], may also can enhance the progress on cancer bioprinting. The production of hybrid scaffolds, by the combination of two different 3D printing technologies [195], may also be helpful to reproduce more of the natural extracellular matrix, which may be applied to rebuild the cancer microenvironment in vitro. Therefore, not only are cancer bioprinting advances necessary, but studies not directly related to cancer bioprinting may also be modified to refine high resolution multi-material bioprinters [196], to accomplish more comprehension on tumor biology and targeted treatments.

## Figures and Tables

**Figure 1 micromachines-12-00764-f001:**
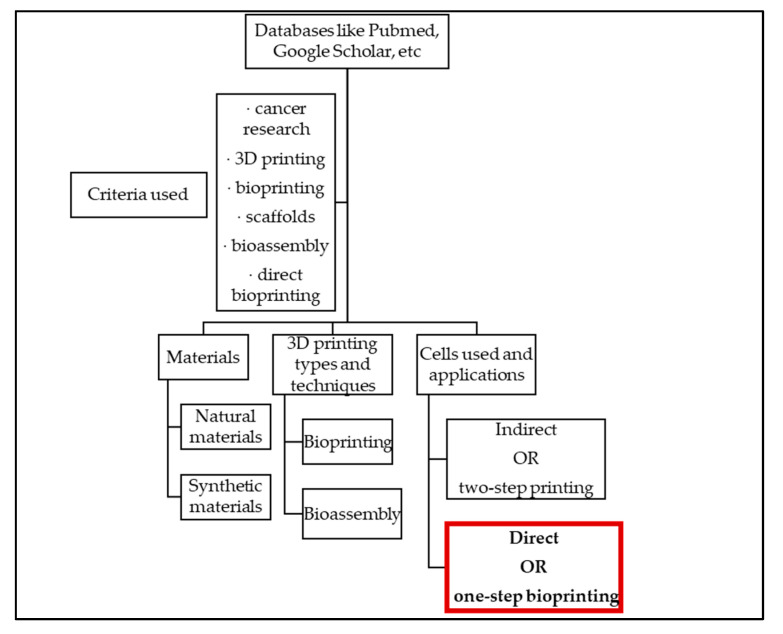
Methodology and criteria used to select the articles, and organization of the review paper. The last section, “cells used and applications”, focuses on direct or one-step bioprinting, using different types of cancer cells.

**Figure 2 micromachines-12-00764-f002:**
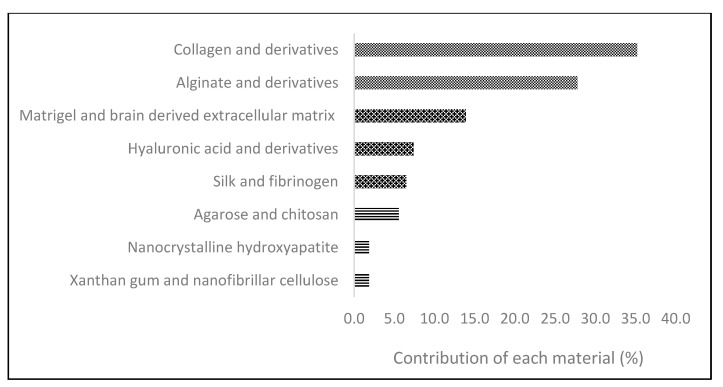
Contribution of each natural biopolymer (%) to the total of natural biopolymers used in 3D bioprinting.

**Figure 3 micromachines-12-00764-f003:**
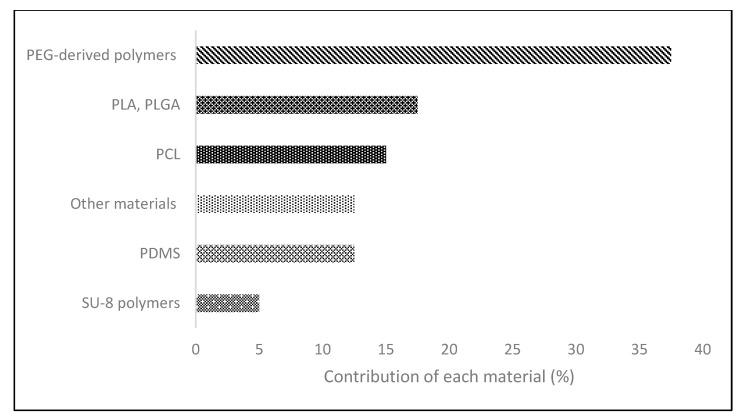
Contribution of each group of synthetic polymers (%) to the total of synthetic polymers. Polyethylene glycol (PEG), polylactic acid (PLA), poly(lactic-co-glycolic) acid (PLGA), polycaprolactone (PCL), and polydimethylsiloxane (PDMS).

**Table 1 micromachines-12-00764-t001:** Description of natural and synthetic materials, their source, advantages and disadvantages of using them.

Material	Source	Advantages	Disadvantages	References
Collagen	Natural, peptide	· Good for cell adhesion · Biocompatible · Low toxicity · Low immunogenicity	· Problems on mechanical strength · Problems on sterilization · Unstable in aqueous conditions	[48,49]
Gelatin	Natural, peptide	· Good cell adhesion and infiltration · Stable at high temperatures · Biodegradable · Non immunogenic	· Low stability · Controversial bioactivity	[50,51]
Alginate	Natural, polysaccharide	· Mimic functions of extracellular matrix · Biocompatible and cytocompatible · Biodegradable and bioabsorbable	· Problems on sterilization · Low cell adherence · Poor mechanical properties	[52,53]
Matrigel	Natural, derived from animal sarcoma	· Mimic more the in vivo microenvironment	· Batch-to-batch variability · Complexity of the composition	[54]
BdECM ^1^	Natural, derived from brain	· Easy to obtain · Tissue specificity	· Potentially immunogenic	[55]
Hyaluronic acid	Natural, polysaccharide	· Non immunogenic · Biocompatible · Osseocompatible	· Fragile · Low biodegradability	[56,57]
Silk	Natural, peptide	· Great strength and elasticity · Biocompatible · Thermostable · Assists on cell migration and vascularization	· Induction of degradation · Possible immunogenicity	[58,59]
Fibrinogen	Natural, peptide	· Biocompatible · Cell-adhesive and binding properties · Non immunogenic	· Poor mechanical strength · High degradation	[60]
Agarose	Natural, polysaccharide	· Great biocompatibility · Non immunogenic · Reversible gelation	· Low cell adhesion · Non-degradable	[61,62]
Chitosan	Natural, polysaccharide	· Promotes cell adhesion · Anti-inflammatory · Non-toxic	· Low mechanical strength · Low solubility · Fast degradation in vivo	[63,64]
Hydroxyapatite-based	Natural, mineral	· Similar chemical and crystallographic structures to human bone · Biocompatible	· Fragile · Low tensile strength	[65]
Xantham gum	Natural, polysaccharide	· Non-toxic · Safe to use	N.A.	[66]
Cellulose-based	Natural, polysaccharide	· Stable structure · Good mechanical properties · Biocompatible and cytocompatible	· Inside the human body, it behaves as non-degradable	[67,68]
PEG ^2^	Synthetic	· Biocompatible · Elastic · Bio adhesive · Non immunogenic	· Insoluble networks · Bioinert origin	[69,70]
PLA ^3^	Synthetic	· Biocompatible and cytocompatible · Good mechanical strength and degradation rate	· Fragile · Hydrophobic	[71]
PLGA ^4^	Synthetic	· Great cell adhesion and proliferation · Good mechanical properties	· Possible biocompatibility issues	[72]
PCL ^5^	Synthetic	· Non-toxic · Cytocompatible · Good mechanical properties · Controls cell proliferation and angiogenesis	· Hydrophobicity · Low bioactivity	[73,74]
PDMS ^6^	Synthetic	· Inert · Non-toxic	· Hydrophobic · Elasticity restrictions	[75]
SU-8 polymers	Synthetic	· Chemical stability · Good mechanical and optical properties	· Restrict adhesion selection	[76]

^1^ Brain-derived extracellular matrix; ^2^ polyethylene glycol; ^3^ polylactic acid; ^4^ poly(lactic-co-glycolic); ^5^ polycaprolactone; ^6^ polydimethylsiloxane.

**Table 2 micromachines-12-00764-t002:** Description of the main materials used, the cost, speed, and some limitations on each type of 3D printing technology described.

	Main Materials	Cost	Speed	Problems	References
Laser printing	Mainly metal powders	Expensive	Fast	Requires post-processing techniques	[86]
Lithography	Resin and photocurable polymers	Expensive	Fast	Possible cytotoxicities	[87]
Inkjet printing	Mainly ceramic powders and thermoplastics	Cheap	Fast	Low mechanical strength	[87]
Droplet-based printing	Mainly photocurable polymers	Cheap	Fast	Low mechanical strength	[86]
Electrospinning	Mainly thermoplastics	Cheap	Fast	Limited control on pores size	[88]
Gas foaming	Polymers	Cheap	-	Limited reproducibility	[89]
Freeze-casting	Mainly metal powders	Cheap	-	Limitations on gas diffusion	[90]

**Table 3 micromachines-12-00764-t003:** Description of the bioinks composition, type of cells, and bioprinting technology used in breast cancer research.

Bioink Composition	Cells Used	3D Bioprinting Technology	Reference
Sodium alginate beads	MDA-MB-231 ^1^	Laser-direct writing	[107]
PED-DMA ^2^ and gelatin type A	MCF-7 ^3^	Extrusion printing (valve-based)	[108]
Alginate-collagen microbeads	MDA-MB-231, MCF-7 and adipose cells	Laser-direct writing	[80]
PEG ^4^ coating–TMSPM ^5^ photoinitiator and ME-GEL ^6^	MDA-MB-231, MCF-7 and MCF-10A ^7^	Photolithography (photomask)	[109]
Collagen	MCF-7	Extrusion printing	[110]
Microfluidic device (PDMS ^8^), collagen type I, and Matrigel	MDA-MB-231, hBM-MSCs ^9^ and HUVEC ^10^	Extrusion printing	[111]
PEG–DEX ^11^ system (not scaffolds)	MDA-MB-231, MCF-7, HepG2 ^12^, HCT-116 ^13^, ES D3 cells ^14^, NIH-3T3 ^15^	Hanging drop (spheroids)	[102]
Sodium alginate	Dil-positive cells (NT, CTSL KD ^16^), MDA-4T1 ^17^	Inkjet printing	[83]
Cell suspension in PBS ^18^	MCF-7	Thermal inkjet printing	[105]
Neutralized rat tail collagen type I	MCF-7, MDA-MB-468 ^19^ and MCF-12A ^20^	Customized Felix 3.0 extrusion printer	[106]
MeHA ^21^, HA ^22^, ME-GEL, gelatin	21 PT ^23^ and ADMSCs ^24^	Extrusion printing	[112]
Matrigel; sodium alginate-gelatin (hydrogel I) and alginate-collagen (hydrogel II)	MDA-MB-231, MCF-7, MCF-10A, MCF-10A-NeuN, breast epithelial cells	Coaxial extrusion printing	[113]
Alginate and gelatin	MCF-7, HCC1143 ^25^, SKBR3 ^26^, MDA-MB-231, HUVEC and fibroblasts	Extrusion printing	[95]
Alginate and gelatin	MDA-MB-231 and IMR-90 ^27^	Extrusion printing	[114]
ME-GEL, nHa, and Irgacure 2959 photoinitiator	MDA-MB-231 and hBM-MSCS	Laser printing	[115]
Peptide-conjugated alginate fibers	MDA-MB-231 and RAW26.7 ^28^	Extrusion printing	[96]
Alginate-gelatin (3:2)	MCF-7 and ADSCs ^29^	Extrusion printing	[36]
Alginate; ADA-GEL ^30^; HA-SH ^31^ and PEGDA ^32^	MDA-MB-231, MCF-7, Mel Im ^33^ and MV3 ^34^	Extrusion printing	[116]

^1^ Claudin-low breast cancer cell line; ^2^ polyethylene glycol dimethacrylate; ^3^ breast cancer cell line; ^4^ polyethylene glycol; ^5^ 3-(trimethoxysilyl)propyl methacrylate photoinitiator; ^6^ methacrylated gelatin; ^7^ non-tumorigenic breast cell line; ^8^ polydimethylsiloxane; ^9^ human bone marrow-mesenchymal stem cells; ^10^ human umbilical vein endothelial cells; ^11^ dextran; ^12^ hepatocellular carcinoma cell line; ^13^ human colorectal carcinoma cell line; ^14^ clonal embryonic stem cells; ^15^ embryonic fibroblast cell line; ^16^ cathepsin L knock down cells; ^17^ epithelial-like breast cancer cell line; ^18^ phosphate-buffered saline; ^19^ breast cancer cell line; ^20^ non-tumorigenic breast cancer cell line; ^21^ methacrylated hyaluronic acid; ^22^ hyaluronic acid; ^23^ human epidermal receptor 2 positive breast primary breast cancer cells; ^24^ adipose-derived mesenchymal stem/stromal cells; ^25^ breast cancer basal-like cell line; ^26^ breast cancer HER2 amplified cell line; ^27^ human fibroblast cell line; ^28^ macrophages; ^29^ adipose-derived stromal cells; ^30^ alginate dialdehyde crosslinked with gelatin; ^31^ hyaluronic acid modified with thiol groups; ^32^ polyethylene glycol diacrylate; ^33^ human melanoma cell line; ^34^ human melanoma cell line.

**Table 4 micromachines-12-00764-t004:** Description of the bioink composition, type of cells, and bioprinting technology used in brain-associated cancer research.

Bioink Composition	Cells Used	3D Bioprinting Technology	Reference
Collagen or BdECM ^1^	U-118 ^2^ and HUVEC ^3^	Extrusion printing	[39]
Alginate	GSC23 ^4^ and U-118	Coaxial extrusion printing	[118]
Gelatin-alginate-fibrinogen (GAF hydrogel)	SU3 ^5^ and U-87 ^6^	Extrusion printing	[120]
Alginate, a microfluidic device of PDMS ^7^ and SU-8 2050 epoxy	U-251 ^8^ and HepG2 ^9^	Inkjet printing	[119]
Direct bioprinting of cells	U-118 GFP ^10^ labeled, GBM4 ^11^, CD1 ^12^, C57BL ^13,^ and Ipsc ^14^-derived hnp cells	Extrusion printing	[104]
Alginate, gelatin, and fibrinogen (GAF hydrogel)	U-118	Extrusion printing	[40]
Gelatin methacryloyl and gelatin	GL261 ^15^, GAMs ^16^ and RAW 264.7 ^17^	Extrusion printing	[79]
Agarose and collagen type I	SH-SY5Y ^18^, UC-MSCs ^19,^ and HUVEC	Droplet printing	[121]
Alginate modified with RGDS ^20^, HA ^21,^ and collagen type I	U-87MG, GSCs ^22^, GASCs ^23^, microglia, WI-38 ^24^ and MM6 cells ^25^	Fab@Home or Renishaw PLC multi-nozzle extrusion printers	[38]
BdECM and silicone	Glioblastoma cells and HUVEC	Extrusion printing	[122]
Collagen type I, III or IV, and thiol-HA ^26^	OSU2 cells ^27^ and astrocytes	Extrusion printing	[123]
Alginate and gelatin (shell) and fibrinogen (core)	GSC23 and hMSCs ^28^	Coaxial extrusion printing	[124]

^1^ Brain-derived extracellular matrix; ^2^ glioblastoma cell line; ^3^ human umbilical vein endothelial cells; ^4^ glioblastoma stem cells; ^5^ glioblastoma stem cells; ^6^ glioblastoma stem cells; ^7^ polydimethylsiloxanes; ^8^ glioblastoma cell line; ^9^ hepatocellular carcinoma cell line; ^10^ green fluorescent protein; ^11^ human brain tumor cell line; ^12^ cells from immunodeficient nude mice; ^13^ cells from inbred mice; ^14^ induced pluripotent stem cells derived from human neural progenitor cells; ^15^ cells from glioma model; ^16^ glioblastoma-associated macrophages; ^17^ cells from Abelson leukemia virus-induced tumor model; ^18^ human bone marrow-derived epithelial-neuroblastoma immortalized cells; ^19^ human primary umbilical cord-derived mesenchymal stromal cells; ^20^ arginylglycylaspartic acid; ^21^ hyaluronic acid; ^22^ glioblastoma stem cells; ^23^ glioblastoma-associated stromal cells; ^24^ human fibroblasts derived from fetal lung tissue; ^25^ monocytes and macrophages from Adult acute monocytic leukemia; ^26^ thiolated hyaluronic acid; ^27^ patient-derived glioblastoma cells; ^28^ human mesenchymal stromal cells.

**Table 5 micromachines-12-00764-t005:** Description of the bioinks composition, type of cells, and bioprinting technology used in lung-associated cancer research.

Bioink Composition	Cells Used	3D Bioprinting Technology	Reference
PEG ^1^-RGDS ^2^, PEG-PQ ^3^ scaffolds	344SQ ^4^, 393P ^5^ and 344P ^6^	Droplet printing (white light polymerization)	[44]
PEG-SVA ^7^, PEG-RGDS, PEG-PQ-PEG ^8^, and microfluidic device of PDMS ^9^	344SQ, HVP ^10^ and HUVEC ^11^	Extrusion printing	[130]
Gelatin-alginate hydrogel	A549 ^12^ and A95D ^13^	Livprint Norm extrusion printer	[129]
Gelatin-sodium alginate-Matrigel hydrogel	A549 and Primary ICC ^14^ cells	SUNP ALPHA-CPT1 Multinozzle extrusion printer	[131]

^1^ polyethylene glycol; ^2^ arginylglycylaspartic acid; ^3^ matrix metalloproteinase (MMP)-sensitive peptide; ^4,5,6^ lung adenocarcinoma cells from mutated mice; ^7^ succinimidyl valerate; ^8^ MMP-2 and -9 sensitive degradation sequence found in the alpha chain of type I collagen; ^9^ polydimethylsiloxanes; ^10^ human vascular pericytes; ^11^ human umbilical vein endothelial cells; ^12^ human alveolar adenocarcinoma cell line; ^13^ lung cancer cell line; ^14^ intrahepatic cholangiocarcinoma cells.

**Table 6 micromachines-12-00764-t006:** Description of the bioink composition, type of cells, and bioprinting technology used in liver-associated cancer research.

Bioink Composition	Cells Used	3D Bioprinting Technology	Reference
Alginate, a microfluidic device of PDMS ^1^ and SU-8 2050 epoxy	HepG2 ^2^ and U-251 ^3^	Inkjet printing	[119]
PEG ^4^–DEX ^5^ system (not scaffolds)	HepG2 and MDA-MB-231 ^6^, MCF-7 ^7^, HCT-116 ^8^, ES D3 cells ^9^, NIH-3T3 ^10^	Hanging drop (spheroids)	[102]
Alginate beads	MHCC97L ^11^ and HCCLM3 ^12^	Extrusion printing	[138]
Rat tail collagen type I	HepG2 and 3T3-J2 ^13^	Hanging drop (spheroids)	[103]

^1^ polydimethylsiloxane; ^2^ hepatocellular carcinoma cell line; ^3^ glioblastoma cell line; ^4^ polyethylene glycol; ^5^ dextrans; ^6^ claudin-low breast cancer cell line; ^7^ breast cancer cell line; ^8^ human colorectal carcinoma cell line; ^9^ clonal embryonic stem cells; ^10^ embryonic fibroblast cell line; ^11,12^ adult hepatocellular carcinoma cell lines; ^13^ subclones of an embryonic fibroblast cell line.

**Table 7 micromachines-12-00764-t007:** Description of the bioinks composition, type of cells, and bioprinting technology used in reproductive-associated cancer research.

Bioink Composition	Cells Used	3D Bioprinting Technology	Reference
Gelatin-alginate-fibrinogen (1:2:1) hydrogel	HeLa ^1^ cells	Extrusion printing	[155]
Matrigel	OVCAR-5 ^2^ and MRC-5 ^3^	Extrusion printing (two extruders)	[156]
Agarose	SkOV3 ^4^	Extrusion printing	[157]
RADA16-I hydrogel	A2780 ^5^, A2780/DDP ^6^ and SkOV3	Extrusion printing	[47]
Matrigel	P19 ^7^ cells	Laser direct writing (MAPLE direct writing)	[154]

^1^ cervical cancer cell line; ^2^ high-grade ovarian serous adenocarcinoma cell line; ^3^ normal human fibroblast cell line; ^4^ ovarian serous cystadenocarcinoma cell line; ^5^ ovarian endometrioid adenocarcinoma cell line; ^6^ ovarian endometrioid adenocarcinomas cisplatin-resistant cell line; ^7^ embryonal carcinoma cell line.

**Table 8 micromachines-12-00764-t008:** Description of the bioink composition, type of cells, and bioprinting technology used in gastric and colorectal-associated cancer research.

Bioink Composition	Cells Used	3D Bioprinting Technology	Reference
Alginate	HCT-116 ^1^ and HCT-116R ^2^	Extrusion printing	[169]
Alginate-hyaluronic acid hydrogel	MKN45 ^3^ and bmMSCs ^4^	Extrusion printing (spheres)	[170]

^1^ human colon carcinoma cell line; ^2^ human 5-fluorouracil-chemoresistant colon carcinoma cell line; ^3^ gastric adenocarcinoma cell line; ^4^ patient-derived bone marrow mesenchymal stem cells.

**Table 9 micromachines-12-00764-t009:** Description of the bioink composition, type of cells, and bioprinting technology used in skin-associated cancer research.

Bioink Composition	Cells Used	3D Bioprinting Technology	Reference
Matrigel:cells (11:1)	MV3dc ^1^	Pneumatic extrusion printing	[179]
Alginate; ADA-GEL ^2^; HA-SH ^3^ and PEGDA ^4^	Mel Im ^5^ and MV3 ^6^, MDA-MB-231 ^7,^ and MCF-7 ^8^	Extrusion printing	[116]

^1^ melanoma cell line modified with plasmid pGL4.23 MCAT-EGFP; ^2^ alginate dialdehyde cross-linked with gelatin; ^3^ hyaluronic acid modified with thiol groups; ^4^ polyethylene glycol diacrylate; ^5^ human melanoma cell line; ^6^ human melanoma cell line; ^7^ claudin-low breast cancer cell line; ^8^ breast cancer cell line.

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
