# Peer review of "Cancer Cell Direct Bioprinting: A Focused Review"

_micromachines, 2021, doi:10.3390/mi12070764_

Round 1

Reviewer 1 Report

In the review „Cancer cell bioprinting: A focused review”, authors summarize experimental approaches based on direct bioprinting for studying cancer cells. Overall, the review is informative and well structured: in particular, the introduction explains that direct and indirect bioprinting are possible techniques to study 3D cell constructs, while the following paragraphs provide a comprehensive classification of several cancer cell lines investigated via direct bioprinting.

Nevertheless, the reviewer has a few suggestions to improve the manuscript:

  1. Since the reviewed articles mainly refer to direct bioprinting techniques as defined by the authors, the title could be re-formulated as “Cancer cell direct bioprinting: A focused review”.
  2. The general procedure for fabricated scaffolds described in lines 37-39 is not always applicable. Indeed, many scaffolds, also obtained via bioprinting techniques, cannot be designed via specific CAD software (e.g., collagen-based ones). Moreover, the deposition of materials is not necessarily uniform. Please reformulate this paragraph, also considering that ref. 10 does not fit to the overall context of the review.
  3. Although the main focus is on direct bioprinting techniques, a brief comparison of the advantages and disadvantages of direct vs. indirect bioprinting might be helpful for the reader. E.g. one should consider that the printing procedure itself might harm the cell survival and harsh conditions during scaffold fabrication might decrease cell survival compared to the indirect procedure, where scaffold fabrication and cell seeding are separated. In this regard, the authors also state that direct bioprinting might be the method yielding higher cell viability, however, without giving any explanation why this might be the case.
  4. The statement that breast cancer is the cancer type with the highest mortality rate for both sexes (54-56) does not coincide with statistics in the cited reference. Moreover, it is questionable if breast cancer is a well-chosen example for the mortality rate of both sexes. Although breast cancer cases are reported in men, they only rarely occur. To avoid misinterpretations, the authors might use another example for male patients or rephrase accordingly.
  5. Although the section on the concept of cancer stem cells is informative, it is very sparse. Given the complexity of these topics, it is not entirely clear how the chosen statements about ALDH1 and EMT contribute to the context of cancer cell bioprinting. In this regard, it might be of higher interest to provide more information about the structural scaffolding properties of the cancer cell niches.
  6. 79: It might be helpful for the reader to briefly provide some information about the mentioned rheological standards, i.e. what are the structural properties of the ECM that one should imitate and what are the main molecules present in the ECM. Given the context that collagen is the most abundant protein in the ECM, it seems logically that most of the later cited studies use collagen as a biomaterial.
  7. In sections 2.1 and 2.2, authors compare percentage amounts of papers referring to natural-derived biopolymers and synthetic polymers (section 2.1), and synthetic polymers (section2.2). It is not really clear how they calculated these fractions and whether these refer to the overall references cited in this review, or, for example, to the results of keyword-based search queries on databases. Please clarify.
  8. 134: The authors state that synthetic polymers have a better reproducibility due to the controlled chemical manufacturing. Maybe they could also briefly highlight the drawbacks of natural biomaterials in this regard?
  9. As the authors correctly point out in section 3 (“3D printing techniques”), bioprinting (either direct or indirect) is only one possible technique to realize 3D cellular constructs. One-two main reviews per each of the also mentioned techniques (e.g., laser writing, lithography, inkjet bioprinting, electrospinning, etc.) could be useful to the reader to eventually compare their advantages/disadvantages with respect to (direct) bioprinting.
  10. In lines 159-164, authors state that only a minority of the reviewed literature is based on scaffold printing followed by cell seeding. This may be true for (direct) bioprinting, but may be a main (or the only) strategy in case of other techniques (e.g., electrospinning). Authors should better highlight this aspect.
  11. The sub-division of the direct bioprinting topic between scaffold-based methods and scaffold-free approaches, which is explained in lines 165-167, can be misleading: indeed, in lines 45-51, indirect bioprinting was already defined as a scaffold-based strategy. Could the authors help to clarify these definitions, eventually with the help of a figure/scheme?
  12. 191: It remains unclear, why spheroids should enable a better cell-cell or cell-matrix communication compared to other techniques. Please specify.
  13. Was a specific criterion followed for the lists 1-7? Are the cited works sorted by cells, or materials? If possible, include a classification within the lists helping the reader to better find his/her material/cell line of interest.
  14. Reference to COVID-19 as a co-factor of high mortality in patients with lung cancer is not necessary in the context of the review and could be removed. The prognosis of several cancers is negatively influenced by a variety of co-morbidities and conditions.
  15. Finally, could the authors shortly comment about the reduced amount of papers regarding gastric/colorectal cancer and skin-associated cancer studies based on direct bioprinting (four references in total vs. eighteen for breast cancer alone)?
  16. General remark: Is it possible to give a brief overview about the advantages/disadvantages (spatial resolution, costs, speed…) of the different 3D bioprinting technologies? Maybe in a table?

Author Response

  1. Since the reviewed articles mainly refer to direct bioprinting techniques as defined by the authors, the title could be re-formulated as “Cancer cell direct bioprinting: A focused review”. The authors accepted the suggestion and changed the title for “Cancer cell direct bioprinting: A focused review”.
  2. The general procedure for fabricated scaffolds described in lines 37-39 is not always Indeed, many scaffolds, also obtained via bioprinting techniques, cannot be designed via specific CAD software (e.g., collagen-based ones). Moreover, the deposition of materials is not necessarily uniform. Please reformulate this paragraph, also considering that ref. 10 does not fit the overall context of the review. The authors checked the text and erase the phrase because it could be misunderstood.  
  3. Although the main focus is on direct bioprinting techniques, a brief comparison of the advantages and disadvantages of direct vs. indirect bioprinting might be helpful for the

E.g. one should consider that the printing procedure itself might harm the cell survival and harsh conditions during scaffold fabrication might decrease cell survival compared to the indirect procedure, where scaffold fabrication and cell seeding are separated. In this regard, the authors also state that direct bioprinting might be the method yielding higher cell viability, however, without giving any explanation why this might be the case. The authors accepted the suggestions and put a table to show graphically the differences between direct and indirect bioprinting.

  1. The statement that breast cancer is the cancer type with the highest mortality rate for both sexes (54-56) does not coincide with statistics in the cited reference. Moreover, it is questionable if breast cancer is a well-chosen example of the mortality rate of both sexes. Although breast cancer cases are reported in men, they only rarely occur. To avoid misinterpretations, the authors might use another example for male patients or rephrase The authors rewrote the paragraph, to avoid confusion.
  2. Although the section on the concept of cancer stem cells is informative, it is very sparse. Given the complexity of these topics, it is not entirely clear how the chosen statements about ALDH1 and EMT contribute to the context of cancer cell bioprinting. In this regard, it might be of higher interest to provide more information about the structural scaffolding properties of the cancer cell niches. The authors erase the incoherent statements, and rewrote the paragraph, providing an introduction for the next paragraph’s explanations.
  3. 79: It might be helpful for the reader to briefly provide some information about the mentioned rheological standards, i.e. what are the structural properties of the ECM that one should imitate and what are the main molecules present in the ECM. Given the context that collagen is the most abundant protein in the ECM, it seems logically that most of the later cited studies use collagen as a biomaterial. The authors rephrased the paragraph to connect it with the previous information.
  4. In sections 2.1 and 2.2, authors compare percentage amounts of papers referring to natural- derived biopolymers and synthetic polymers (section 2.1), and synthetic polymers (section2.2). It is not really clear how they calculated these fractions and whether these refer to the overall references cited in this review, or, for example, to the results of keyword-based search queries on databases. Please clarify. The authors revised the section and clarified how the articles have been selected in this review paper.
  5. 134: The authors state that synthetic polymers have a better reproducibility due to the controlled chemical manufacturing. Maybe they could also briefly highlight the drawbacks of natural biomaterials in this regard? The authors accepted the suggestions and rewrote the paragraph.
  6. As the authors correctly point out in section 3 (“3D printing techniques”), bioprinting (either direct or indirect) is only one possible technique to realize 3D cellular constructs. One-two main reviews per each of the also mentioned techniques (e.g., laser writing, lithography, inkjet bioprinting, electrospinning, etc.) could be useful to the reader to eventually compare their advantages/disadvantages with respect to (direct) bioprinting. The authors accepted the suggestions and rewrote the paragraph.
  7. In lines 159-164, the authors state that only a minority of the reviewed literature is based on scaffold printing followed by cell seeding. This may be true for (direct) bioprinting but may be a main (or the only) strategy in case of other techniques (e.g., electrospinning). Authors should better highlight this The authors accepted the suggestions and rewrote the entire paragraph.
  8. The sub-division of the direct bioprinting topic between scaffold-based methods and scaffold- free approaches, which is explained in lines 165-167, can be misleading: indeed, in lines 45- 51, indirect bioprinting was already defined as a scaffold-based strategy. Could the authors help to clarify these definitions, eventually with the help of a figure/scheme? The authors accepted the suggestions and rephrase the paragraph to avoid misunderstandings.
  9. 191: It remains unclear, why spheroids should enable a better cell-cell or cell-matrix communication compared to other Please specify. The authors accepted the suggestions and rewrote this section.
  10. Was a specific criterion followed for the lists 1-7? Are the cited works sorted by cells, or materials? If possible, include a classification within the lists helping the reader to better find his/her material/cell line of The authors accepted the suggestions and specified the criteria used to classify the different cancer cells.
  11. Reference to COVID-19 as a co-factor of high mortality in patients with lung cancer is not necessary in the context of the review and could be removed. The prognosis of several cancers is negatively influenced by a variety of co-morbidities and The authors accepted the suggestions and erase the content to avoid confusion.
  12. Finally, could the authors shortly comment about the reduced amount of papers regarding gastric/colorectal cancer and skin-associated cancer studies based on direct bioprinting (four references in total vs. eighteen for breast cancer alone)? The authors accepted the suggestions and gave a short comment/opinion on the topic.
  13. General remark: Is it possible to give a brief overview about the advantages/disadvantages (spatial resolution, costs, speed…) of the different 3D bioprinting technologies? Maybe in a table? The authors accepted the suggestions put a table to show the different 3D printing technologies mentioned in the text, and their differences and problems.

Reviewer 2 Report

Introduction:

3D printing is not the same as rapid prototyping. 3D printing is often used interchangeably with 'additive manufacturing' – the process of building structures from the ‘bottom-up’ using computer aided design and manufacturing (CAD/CAM). Rapid prototyping is a broader term, including techniques such as 3D printing, but also laser cutting, machining, etc. Other used terminology referring to 3D printing, scaffold building and biomedical engineering in general should be described with more care. Please be careful when using the terminology, there is already a lot of confusion in regards to that. For definitions I recommend the authors to refer to the publications by Groll et al: https://doi.org/10.1088/1758-5090/8/1/013001, as well as Ramos & Moroni: https://doi.org/10.1089/ten.tec.2019.0344. Also, to get a better overview over the different approaches and techniques, see some additional review papers on 3D bioprinting (e.g. Ng et al., 2019; Vijayavenkataraman et al., 2018; etc).

It is not clear from the text what ‘one-step’ or ‘direct bioprinting’ and what ‘two-step’ or ‘indirect printing’ are. What type of printing does this refer to, is it just for extrusion based techniques, or also SLA, LIFT, etc.?

The structure of the introduction should be improved. The section on cancer seems out of place and it is not clear how it relates to 3D bioprinting, or how 3D bioprinting can help solve cancer. If this is the theme of the manuscript also include this more in the abstract.

The last section of the introduction has the same issues as the first one. Asserting that hydrogels are THE BEST bioprinting materials is misleading and inaccurate. Please be more thorough and specific. Murphy & Atala (ref 9) give a very good description on the requirements of scaffold materials in 3D bioprinting.

Materials:

The first few sentences in this section are too general and can be misunderstood. ECM function is tissue specific, also epithelia have little to no ECM at all. Rephrasing that would help. The cited material selection process (Ashby) was not developed for biological use cases. While this reviewer agrees that the provided information is relevant, it should be clarified how and on which use-cases the described process is built. The biological aspects of material requirements should also be described in more detail. Which properties and why are they important?

In the section on the polymer types (natural and synthetic) more details and perhaps a reworked text structure would improve the content greatly. How are the mentioned polymers different from each other in terms of composition, source, properties, advantages/disadvantages, etc.? In line 109 the authors mention that alginate is the most commonly used natural polymer. Most commonly used for what purpose? Why is it in second place behind collagen in the graphic? What is the value of grouping the polymers in the graphics? The authors describe percentages of polymer occurrence in literature. How exactly did the authors obtain these numbers? What was the literature-review methodology?

3D printing techniques:

Tests on animals ARE a part of preclinical studies. What kind of gap is there? Or do the authors mean, that tissue engineering (not bioprinting per se) presents a possible alternative for animal models? The sentence on extrusion and viscosity seems unrelated to the previous sentence. It is also not quite true. For some extrusion based techniques, viscosity is not the main limiting factor (e.g. support bath printing). Also foaming and freeze-casting can hardly be considered bioprinting. The other techniques need some more information.

It is the opinion of this reviewer, that all bioprinting techniques are “scaffold-based” technologies, as they require some sort of ‘binder’ to print cellular components. If organoids or even cells in general are deposited without some scaffold material, the process is called bio-assembly, which can together with bioprinting be considered as a part of the broader term biofabrication.

Cellular classification:

The authors name a few case studies used with respective tissue types. What would significantly improve this section would be a bit more detailed descriptions on the individual approaches, their advantages, disadvantages and outcomes. What are the challenges for tissue engineering of respective cancer types?

Specific remarks:

Line 30 – SLA was the first 3D printing technology, not the first 3D technique in general

Line 31 – UV is not the only spectrum compatible with SLA

Line 35 – needs citation

Line 125 – HUEVC should be HUVEC

Line 152 – animal experimentation IS part of preclinical studies

Line 208 – second MOST COMMON cause of death?

Author Response

Introduction:

3D printing is not the same as rapid prototyping. 3D printing is often used interchangeably with 'additive manufacturing' – the process of building structures from the ‘bottom-up’ using computer aided design and manufacturing (CAD/CAM). Rapid prototyping is a broader term, including techniques such as 3D printing, but also laser cutting, machining, etc. Other used terminology referring to 3D printing, scaffold building and biomedical engineering in general should be described with more care. Please be careful when using the terminology, there is already a lot of confusion in regards to that. For definitions I recommend the authors to refer to the publications by Groll et al: https://doi.org/10.1088/1758-5090/8/1/013001, as well as Ramos & Moroni: https://doi.org/10.1089/ten.tec.2019.0344. Also, to get a better overview over the different approaches and techniques, see some additional review papers on 3D bioprinting (e.g. Ng et al., 2019; Vijayavenkataraman et al., 2018; etc). The authors revised all the text and corrected all the misunderstandings of the terminology employed.

It is not clear from the text what ‘one-step’ or ‘direct bioprinting’ and what ‘two-step’ or ‘indirect printing’ are. What type of printing does this refer to, is it just for extrusion based techniques, or also SLA, LIFT, etc.? The authors accepted the comments and rephrase the paragraph to make it easy to understand.

The structure of the introduction should be improved. The section on cancer seems out of place and it is not clear how it relates to 3D bioprinting, or how 3D bioprinting can help solve cancer. If this is the theme of the manuscript also include this more in the abstract. The authors accepted the suggestions and rewrote the paragraph accordingly.

The last section of the introduction has the same issues as the first one. Asserting that hydrogels are THE BEST bioprinting materials is misleading and inaccurate. Please be more thorough and specific. Murphy & Atala (ref 9) give a very good description on the requirements of scaffold materials in 3D bioprinting. The authors accepted the suggestions and rewrote the paragraph accordingly.

Materials:

The first few sentences in this section are too general and can be misunderstood. ECM function is tissue specific, also epithelia have little to no ECM at all. Rephrasing that would help. The cited material selection process (Ashby) was not developed for biological use cases. While this reviewer agrees that the provided information is relevant, it should be clarified how and on which use-cases the described process is built. The biological aspects of material requirements should also be described in more detail. Which properties and why are they important? The authors accepted the suggestions and rewrote the paragraph to clarify that Ashby’s methodology can be adapted to biological uses, on direct bioprinting.

In the section on the polymer types (natural and synthetic) more details and perhaps a reworked text structure would improve the content greatly. How are the mentioned polymers different from each other in terms of composition, source, properties, advantages/disadvantages, etc.? In line 109 the authors mention that alginate is the most commonly used natural polymer. Most commonly used for what purpose? Why is it in second place behind collagen in the graphic? What is the value of grouping the polymers in the graphics? The authors describe percentages of polymer occurrence in literature. How exactly did the authors obtain these numbers? What was the literature-review methodology? The authors already explained the methodology used for the selection of the articles employed in the review paper.

3D printing techniques:

Tests on animals ARE a part of preclinical studies. What kind of gap is there? Or do the authors mean, that tissue engineering (not bioprinting per se) presents a possible alternative for animal models? The sentence on extrusion and viscosity seems unrelated to the previous sentence. It is also not quite true. For some extrusion based techniques, viscosity is not the main limiting factor (e.g. support bath printing). Also foaming and freeze-casting can hardly be considered bioprinting. The other techniques need some more information. The authors already introduced more information on each type of 3D printing technology, in a table.

It is the opinion of this reviewer, that all bioprinting techniques are “scaffold-based” technologies, as they require some sort of ‘binder’ to print cellular components. If organoids or even cells in general are deposited without some scaffold material, the process is called bio-assembly, which can together with bioprinting be considered as a part of the broader term biofabrication. The authors accepted the opinion of the reviewer and rewrote the paragraph on “scaffold-free” technologies.

Cellular classification:

The authors name a few case studies used with respective tissue types. What would significantly improve this section would be a bit more detailed descriptions on the individual approaches, their advantages, disadvantages, and outcomes. What are the challenges for tissue engineering of respective cancer types? The authors accepted the suggestions and put a general introduction on this section to clarify possible confusion. Also, the authors corrected the specific remarks.

Reviewer 3 Report

Although the amount of literature review is extensive, the article needs substantial improvement. There were several major issues with the work, namely:

  • The value of the review paper. Some references were of very general scope (e.g., line 89) and hardly related to the field of tissue engineering. It is unclear if the authors used unbiased methods to collect the sample of articles used to calculate the statistics. If the sample is biased, the value of the calculations given is minimal. In reviewing the various cancer models, some areas were ignored without explanation, such as testicular cancer in the germline cancer models (among others). The bulk of the text consists of simply retelling of the findings in the references, with minimal analysis of the inter-relation of the results, thus the novelty and utility of this review article raises doubts.
  • The lack of depth in the analysis. There is minimal analysis of the processes happening in the engineered tissue and the mechanisms that differentiate a suitable cancer model from a poor one. Although many methods are presented, the key biochemical differences in the end results are overlooked.
  • Factual errors in the text. These need to be fixed, as some statements (noted below) are misleading or simply wrong.

Some specific issues to note:

Line 109 – factually incorrect. The authors’ own data shows that the most common natural polymer used is collagen.

Line 209 – factually incorrect, ref. https://ascopost.com/news/december-2020/globocan-2020-database-provides-latest-global-data-on-cancer-burden-cancer-deaths/

Line 360 – inaccurate; the cervix is not part of the germline.

Line 405 – teratocarcinoma cells are not germline cells.

Lastly, extensive language editing is necessary, especially in some sections, such as the introduction. In some cases, it was hard to understand what was being written.

Author Response

  • The value of the review paper. Some references were of very general scope (e.g., line 89) and hardly related to the field of tissue engineering. It is unclear if the authors used unbiased methods to collect the sample of articles used to calculate the statistics. If the sample is biased, the value of the calculations given is minimal. In reviewing the various cancer models, some areas were ignored without explanation, such as testicular cancer in the germline cancer models (among others). The bulk of the text consists of simply retelling of the findings in the references, with minimal analysis of the inter-relation of the results, thus the novelty and utility of this review article raises doubts. The authors already solved these problems by defining the method used to select the articles of the review paper.
  • The lack of depth in the analysis. There is minimal analysis of the processes happening in the engineered tissue and the mechanisms that differentiate a suitable cancer model from a poor one. Although many methods are presented, the key biochemical differences in the end results are
  • Factual errors in the text. These need to be fixed, as some statements (noted below) are misleading or simply The authors accepted the suggestions and corrected all the inconsistencies in the text. The authors also solved the specific issues.

Round 2

Reviewer 2 Report

I suggest the authors clarify, that they define 'one-step' and 'two-step' printing for the purposes of this manuscript. The cited article by Hollister, 2005 does not define the terms as such. It is also not correct that all bioprinting processes can take place using bioinks with living cells. Think about SLS.

Try avoiding terms like ‘endless possibilities’. Be concrete and be concise.

I don’t understand Table 1 and it’s value here. How were the parameter (e.g. production time, cell integrity, etc.) scores evaluated?

Hydrogels is a very broad group of materials that are prepared by soaking polymers, however, their properties can be very different depending on the nature of each polymer. It is not ‘one promising biomaterial for bioprinting’ as the authors describe.

Lines 241, 243 and a few others require references.

The last paragraph before section 2.1. doesn’t provide significant insight into the methodology of this review. It should be revised or skipped entirely.

The sections on materials, 3D bioprinting techniques and cells have not improved significantly. The questions I had previously were not answered and call into question whether the authors have indeed performed a broad literature review.

Author Response

Reviewer 2 (Report Round 2)

I suggest the authors clarify, that they define 'one-step' and 'two-step' printing for the purposes of this manuscript. The cited article by Hollister, 2005 does not define the terms as such. It is also not correct that all bioprinting processes can take place using bioinks with living cells. Think about SLS. The authors have revised the concepts and clarified them.

Try avoiding terms like ‘endless possibilities’. Be concrete and be concise. The authors corrected the vague terms employed.

I don’t understand Table 1 and its value here. How were the parameter (e.g. production time, cell integrity, etc.) scores evaluated? Because the authors have rephrased the concepts of “one-step” and “two-step” bioprinting, Table 1 is useless and confusing for the readers, so has been erased. 

Hydrogels is a very broad group of materials that are prepared by soaking polymers; however, their properties can be very different depending on the nature of each polymer. It is not ‘one promising biomaterial for bioprinting’ as the authors describe. The authors have rephrased the sentences, avoiding unclear terms.

Lines 241, 243 and a few others require references. The authors have revised all the information, and each sentence has its references.

The last paragraph before section 2.1. doesn’t provide significant insight into the methodology of this review. It should be revised or skipped entirely. The authors have revised the paragraph and rephrased it. Also, they added a figure to clarify the process of selection employed.

The sections on materials, 3D bioprinting techniques and cells have not improved significantly. The questions I had previously were not answered and call into question whether the authors have indeed performed a broad literature review. According to the comments on Round 1, the authors added a Table with information on the source, advantages, and disadvantages of using each material previously mentioned.

Reviewer 3 Report

After revision, the text is significantly improved.

However, I have two more comments:

  1. Table 1 is informative, but the adjectives used are confusing. What is “high” or “low” time? Perhaps there could be more references, or a more detailed explanation what determines this judgement?
  2. I believe there could be more references regarding the cells used in urinary bladder and gastric/colorectal cancer bioprinting, or at least these parts warrant some speculation on why little research has been done in these fields. As of now, these subsections are relatively scarce on information.

Author Response

Reviewer 3 (Report Round 2)

After revision, the text is significantly improved.

However, I have two more comments:

  1. Table 1 is informative, but the adjectives used are confusing. What is “high” or “low” time? Perhaps there could be more references, or a more detailed explanation what determines this judgement? The authors have erased Table 1, because it was confusing for the readers.
  2. I believe there could be more references regarding the cells used in urinary bladder and gastric/colorectal cancer bioprinting, or at least these parts warrant some speculation on why little research has been done in these fields. As of now, these subsections are relatively scarce on information. The authors have been included more information on why there is little research on direct bioprinting of urinary bladder, gastric and colorectal cancers.